# Sublingual Immunotherapy for Japanese Cedar Pollinosis: Current Clinical and Research Status

**DOI:** 10.3390/pathogens11111313

**Published:** 2022-11-09

**Authors:** Daiju Sakurai, Hiroki Ishii, Ayumi Shimamura, Daisuke Watanabe, Takaaki Yonaga, Tomokazu Matsuoka

**Affiliations:** Department of Otorhinolaryngology, Head and Neck Surgery, Interdisciplinary Graduate School of Medicine, University of Yamanashi, 1110 Shimokato, Chuo 409-3898, Yamanashi, Japan

**Keywords:** allergic rhinitis, allergen immunotherapy, Japanese cedar pollen, pollinosis, sublingual immunotherapy

## Abstract

The incidence of Japanese cedar pollinosis is increasing significantly in Japan, and a recent survey suggested that about 40% of the population will develop this disease. However, spontaneous remission is rare. The increased incident rate of Japanese cedar pollinosis is a huge issue in Japan. Allergen immunotherapy is the only fundamental treatment that modifies the natural course of allergic rhinitis and provides long-term remission that cannot be induced by general drug therapy. Sublingual immunotherapy for Japanese cedar pollinosis has been developed and has been covered by health insurance since 2014 in Japan. The indication for children was expanded in 2018. Clinical trials of sublingual immunotherapy for Japanese cedar pollinosis have demonstrated its long-term efficacy and safety. It is recommended for patients who wish to undergo fundamental treatment regardless of the severity of the practical guidelines for the management of allergic rhinitis in Japan. For sublingual immunotherapy, a long-term treatment period of 3 years or longer is recommended to obtain stable therapeutic effects. In recent years, evidence based on basic research and clinical trials has demonstrated sublingual immunotherapy-induced immunological changes and efficacy in patients; however, biomarkers that objectively predict and judge these therapeutic effects need to be established.

## 1. Introduction

Japanese cedar (JC) (*Cryptomeria japonica*) pollinosis is the most prevalent allergic rhinitis (AR) in Japan. The number of patients with JC pollinosis has markedly increased over the last two decades; however, spontaneous remission is rare [1,2,3,4,5]. Therefore, once JC pollinosis develops, many patients may suffer from nasal and ocular allergic symptoms for a long time. The scattering period of JC pollen is long, and the amount of JC pollen scattered has increased in recent years; thus, the symptoms tend to be robust and exacerbated. Symptomatic drug therapy is the main treatment for AR in Japan and no effective preventive treatment for AR has been established to date. However, allergen immunotherapy (AIT) is the only fundamental treatment expected to modify the natural course of AR. Sublingual immunotherapy (SLIT) for patients with AR induced by JC pollen and house dust mites (HDM) has been started as a medical treatment and is covered by health insurance in Japan. SLIT for JC pollinosis was recently demonstrated to be effective and tolerable in clinical trials with a high level of evidence. Although AIT requires long-term treatment of 3 years or more to obtain a stable therapeutic effect, it lacks efficacy in some patients. Recently, immunological changes induced by AIT have been demonstrated in clinical trials or basic research. However, biomarkers that objectively predict and judge therapeutic effects have not been established to date. In this paper, the characteristics of JC pollinosis and the current clinical and research status of SLIT for JC pollinosis are described based on recent findings.

## 2. Characteristics of JC Pollinosis in Japan

JC pollinosis, AR induced by JC pollen (Figure 1), which is frequently accompanied by conjunctivitis, was first reported in the 1960s by Saito et al. [6]. Since the 1950s, JC trees have been planted across Japan in areas other than Hokkaido and Okinawa because of the rising demand for cedar wood. Many of the planted JC trees are now over 30 years old and have high pollen-producing ability, leading to increasing amounts of scattered JC pollen. JC trees disperse a large amount of pollen, over a long period, mainly from February to March, reaching distances over 100 km, and therefore, many patients with JC pollinosis can be found in urban areas such as Tokyo (1). When a large amount of JC pollen scatters it causes severe allergic nasal symptoms, and that these symptoms are exacerbated with repeated exposure has been confirmed by experiments using pollen exposure chambers in Japan [7].

Of note, many cases of JC pollinosis and cypress pollinosis coexist. The major allergens of JC pollen and cypress pollen are highly homologous, and about 70% of JC pollinosis cases are also sensitized to cypress pollen [8,9]. Cypress has been planted in a wide area that stretches between east and west Japan. Figure 2 shows the JC and Cypress pollen counts in Yamanashi Prefecture, which has one of the highest frequencies of patients with JC pollinosis in Japan. The cypress pollen dispersal season continues after the cedar pollen dispersal season. Therefore, many patients with coexisting JC pollinosis and cypress pollinosis have allergic symptoms for a long period from February to April or May.

A questionnaire survey for otolaryngologists and their families in Japan conducted in 2019 reported that the overall prevalence of AR was 49.2%. Thus, approximately one in two Japanese people have AR to an allergen. Compared with similar surveys, the prevalence of JC pollinosis has increased rapidly from 16.2% in 1998 to 26.5% in 2008 and 38.8% in 2019. In a 2019 survey, 30.1% of those aged 5 to 9 years and 49.5% of those aged 10 to 19 years had JC pollinosis, indicating the age of onset is decreasing [1,2,3]. The sensitization rate of JC pollen and the prevalence of JC pollinosis are associated with the amount of JC pollen dispersal [10]. Although the symptoms of JC pollinosis disappear in some cases, their frequency is low in long-term observation studies [4,5].

## 3. General Drug Treatment for JC Pollinosis

Treatments for the symptoms of pollinosis are provided according to the Nasal Allergy Guidelines in Japan. Pharmacotherapy such as second-generation antihistamines are used for patients with mild and moderate symptoms consisting of sneezing and a runny nose. Leukotriene receptor antagonists or prostaglandin D2/thromboxane A2 receptor antagonists are used for moderate symptoms with nasal congestion, and a combination of nasal topical steroids and chemical mediator receptor antagonist such as second-generation antihistamines are recommended for severe cases in Japan [1]. Additionally, early primary pharmacotherapy when pollen scattering starts and symptoms appear is recommended for cases with annual severe symptoms of AR caused by pollen. In Japan, omalizumab, an anti-IgE therapy, has been approved for use in patients aged over 12 years with severe nasal allergic symptoms induced by JC pollinosis that are uncontrolled even with the above drug combinations. The efficacy of add-on omalizumab in patients treated with nasal steroids and antihistamines during the JC pollen season was confirmed in a clinical trial of patients with JC pollinosis [11].

## 4. AIT for JC Pollinosis

Unlike general drug therapies, SLIT, together with conventional subcutaneous immunotherapy (SCIT), is a therapeutic method that is expected to induce long-term remission. Because there is no established method to prevent AR, AIT is in a significant position as a fundamental treatment [1]. SCIT was started in the 1960s in Japan, and SLIT for JC pollinosis as a liquid formulation, Cedartolen^®^, was approved in 2014 as the first SLIT medication in Japan. The liquid formulation of JC pollen SLIT contains a maintenance dose of 2000 JAU of JC pollen extract. Then, a SLIT tablet, Cedarcure^®^, for JC pollinosis with an increased effective concentration was developed and approved in 2018. The JC pollen SLIT tablets used as a maintenance dose contain 5000 JAU of JC pollen extract (10,000 JAU/mL contains 7.3–21 μg/mL of Cry j 1, the major allergen of JC pollen). To date, only one company has developed SLIT for JC pollinosis. The liquid formulation Cedartolen^®^ is now discontinued and only the tablet Cedarcure^®^ is available. There have been no reports of clinical trials of other SLIT preparations for JC pollinosis. Cedarcure^®^ is used at 2000 JAU for one week as a dose-up period, followed by a daily maintenance dose of 5000 JAU.

A tablet of JC pollen SLIT maintained in the oral cavity for one minute disintegrates quickly and is absorbed through the oral floor mucosa. Mild side effects such as local itching and edema are often observed during the dose increase period or the first month after administration. Systemic adverse effects such as anaphylaxis are very rare, and its safety profile is higher than that of conventional SCIT. Currently, a treatment period of 3 years or longer is recommended to obtain a stable and long-term therapeutic effect according to the Nasal Allergy Guidelines of Japan [1]. As for the treatment costs, it is covered by health insurance in Japan; however, in the case of adults, it costs about JPY 100,000 for three years of treatment.

Regarding the indications of AIT for JC pollinosis, AIT modifies the natural course of AR and is a treatment that is expected to lead to remission; thus, it is recommended for patients who desire a fundamental improvement. The Nasal Allergy Guidelines of Japan recommend AIT for AR of any severity, from mild to severe. It is especially recommended for patients whose symptoms are poorly controlled by general drug therapy, patients who experience side effects from drug therapy, and patients who want to reduce drug use [1]. The indication for JC pollen SLIT tablets was expanded to children in 2018, and it is a good indication for children who take it adequately.

Because of the recent increase in JC pollinosis in Japan, the number of patients with concurrent AR caused by JC pollen and HDM has increased. Although there have been no reports on the appropriate method of the concomitant use of sublingual immunotherapy, a clinical trial was conducted on the combined use of JC pollen SLIT and HDM SLIT. There was no difference in safety between the combined treatment and single treatments when they were started to be administered 4 weeks apart [12,13]. Combined therapy is recommended for patients with AR who have symptoms induced by JC pollen and HDM allergens and who wish to be treated simultaneously.

## 5. Clinical Effects of JC Pollen SLIT for JC Pollinosis

Clinical trials of SLIT for patients with JC pollinosis conducted in Japan reported its therapeutic efficacy and safety. Initially, a liquid formulation of standardized JC pollen extract was developed as a SLIT for JC pollinosis. A placebo-controlled, double-blind comparative study of JC pollen SLIT extract was conducted in 531 patients with JC pollinosis aged 12–64 years for two consecutive pollen seasons. In the active group, 2000 JAU of JC pollen extract was used as a maintenance dose. Total nasal symptom medication scores at the peak of symptoms were significantly lower in the active group than in the placebo group, with an 18% reduction in the first season and a 30% reduction in the second season (Table 1). Total ocular symptom medication scores in the active groups were also significantly lower compared with placebo groups, demonstrating the effectiveness of the liquid formulation of JC pollen SLIT extract for nasal and ocular symptoms [14].

Subsequently, a randomized, placebo-controlled, double-blind comparative study was conducted using JC pollen SLIT tablets in 1042 patients with JC pollinosis aged 5–64 years. First, to identify the optimal dose of the JC pollen SLIT tablet, enrolled patients were randomized into four groups receiving a SLIT tablet with 2000, 5000, or 10,000 JAU or placebo for 15 months. As a result, the optimal dose of the JC pollen SLIT tablet was shown to be 5000 JAU, with good efficacy and safety over a 3-year treatment period [15]. Second, patients receiving placebo and the optimal dose (5000 JAU) were then randomized to receive placebo or a SLIT tablet of 5000 JAU for an additional 18 months [16]. Improvements in nasal and ocular symptoms depending on the treatment period of JC pollen SLIT were confirmed. Third, the therapeutic efficacy of JC pollen SLIT tablets was evaluated during the peak symptom period of each JC pollen season over 3 years and 2 years of observation after treatment cessation [17]. The JC pollen SLIT continuation group had the best symptom improvement assessed by the total nasal symptom and medication scores during the 3-year treatment period and the subsequent 2-year follow-up period, showing symptom reduction effects of 46.3% in the third season of the treatment period, 45.3% in the fourth season, and 34.0% in the fifth season of the 2-year follow-up period compared with the placebo group. This indicated that JC pollen SLIT for 3 years maintained disease-modifying effects up to 2 years after the completion of treatment (Table 1). The 1st year efficacy of tablets (5000 JAU) was comparable to the 2nd year efficacy of previous liquid formulations (2000 JAU). Serious adverse events that could irrefutably be attributed to the JC pollen SLIT drug were not observed in these studies.

In the US, Canada, and Europe, Phleum pratense SLIT tablet (Grazax^®^) and Five-grass SLIT tablet (ORALAIR^®^) are approved for allergic rhinitis caused by grass pollen. Randomized, placebo-controlled, double-blind comparative studies were conducted using these grass pollen SLIT tablets, and their clinical effects were investigated during the 3-year treatment period and the subsequent 2-year follow-up period. The improvement rates of the combined score for the grass pollen SLIT tablets (Grazax^®^ and ORALAIR^®^) versus placebo were 36% and 38% at 3 years of treatment and 27% and 28% at 2 years after treatment cessation, respectively [18,19]. JC pollen SLIT showed improvement rates equal to or greater than those of grass pollen SLIT. Regarding the long-term effects of SLIT, grass pollen SLIT and house dust mite SLIT for elderly patients have been shown to be effective up to 5 years after the treatment cessation [20].

In addition, the JC pollen SLIT group showed a significant improvement in nasal and ocular symptoms during the cypress pollen dispersal period following the JC pollen dispersal period compared with the placebo group [21]. The JC pollen extract used for JC pollen SLIT contains the major antigens Cry j 1 and Cry j 2, which have high amino acid sequence homology with Cha o 1 and Cha o 2, which are the major allergens of Japanese cypress pollen [22,23].

JC pollen SCIT has been conducted in Japan for a long time; however, a placebo-controlled double-blind comparative study has not been conducted. According to previous reports based on clinical use in Japan, the efficacy rates of JC pollen SCIT were about 70% [24] (A4). It has been suggested that SCIT may be more effective than SLIT, however, there are no clinical trials comparing the clinical effects of JC pollen SCIT and SLIT.

**Table 1 pathogens-11-01313-t001:** Improvement rate of total nasal symptom medication scores during the peak symptom period in the JC pollen dispersal season in the active group with JC pollen SLIT drop and SLIT tablet vs. the placebo group [14,17].

	During Treatment			After Treatment	
Year	1st	2nd	3rd	4th	5th
JC pollen SLIT drop	18% **	30% **			
JC pollen SLIT tablet	32.1% *	45.1% *	46.3% *	45.3% *	34.0% **

* *p* < 0.001, ** *p* < 0.0001.

## 6. Mechanisms and Factors Related to the Clinical Efficacy of SLIT

Recent clinical trials and basic research reported immunological factors such as cytokines and immune cells fluctuate after AIT, and the mechanisms involved have been studied (Figure 3). When a large amount of allergen is administered under the tongue, it is taken up by dendritic cells (DCs) in the oral mucosa. This leads to the enhanced expression of inhibitory molecules of DCs and production of inhibitory cytokines such as IL-10 and TGF-β. These cytokines promote Th1 immunity and induce regulatory T cells (Treg), which also produce IL-10 and TGF-β. These cytokines suppress the induction and activation of Th2 cells. Follicular helper T cells (Tfh) cells and IL-10-producing regulatory B cells are also induced, the class switching of B cells to IgE antibodies is suppressed, and the production of specific IgG4 and IgA antibodies is induced. Through these actions, the activation of mast cells and eosinophils is suppressed, and allergic symptoms are suppressed at the time of allergen exposure [1,25,26,27]. Additionally, it was recently reported that innate immunity is involved in the pathogenesis of allergic diseases, and that type 2 innate lymphocytes (ILC2) were elevated in the peripheral blood of patients with AR [28]. In addition, AIT decreased the numbers of ILC2 and increased regulatory ILC2 [28,29,30]. This might be associated with the suppression of Th2 cell functions and improvement in the symptoms of AR. However, because ILC do not have antigen specificity, the mechanism of their involvement in AIT is not fully understood.

The relationship between the therapeutic efficacy and baseline data of patients who participated in clinical trials of JC pollen SLIT has been examined [31]. It was observed that treatment efficacy was lower when the body mass index was 25 or higher. However, no association was observed between clinical efficacy and other baseline factors including disease duration, sensitization to other allergens, total IgE, JC pollen-specific IgE, or the specific IgE/total IgE ratio. Serum JC pollen-specific IgE and specific IgG4 were elevated after the initiation of JC pollen SLIT, but were not associated with therapeutic efficacy. Furthermore, the severity of symptoms in the 1st season of JC-pollen SLIT did not predict that in the second season.

It has not been established a biomarker to evaluate the therapeutic effects of SLIT objectively, the evaluation has been performed based on changes in symptoms. Immunological parameters such as specific IgE and specific IgG4 have been investigated so far, however, the relationship with clinical efficacy has not been shown. On the other hand, SLIT has been shown to have a high placebo effect in clinical trials, and the establishment of an objective parameter is expected.

Allergen-specific Th2 cells are thought to have a central role in the pathogenesis of allergic diseases including AR. The change in JC pollen-reactive Th2 cells in JC pollen SLIT has been investigated [32]. Peripheral blood mononuclear cells from 40 patients who participated in a phase III trial of JC pollen SLIT for JC pollinosis [14] were stimulated with JC pollen allergen, cultured, and the numbers of JC pollen-reactive Th2 cells that produced type 2 cytokines such as IL-4 and IL-5 were measured. After the initiation of JC pollen SLIT, the placebo group and non-responders in the active group showed a significant increase in JC pollen-reactive Th2 cells during the JC pollen dispersal period. In contrast, the increase in JC pollen-reactive Th2 cells was suppressed in responders of the active group, and the frequencies of JC pollen-reactive Th2 cells were significantly lower compared with those of the placebo and non-responder active groups. In addition, a significant increase in JC pollen-reactive regulatory T cells was confirmed only in the responders in the active SLIT group after the start of JC pollen SLIT.

Recently, it was shown that ST2-positive pathogenic memory Th2 cells, which produce large amounts of IL-5 in response to antigens, have an important role in the pathogenesis of allergic diseases [33,34]. IL-5 production was increased in Th2 cells from patients with JC pollinosis upon JC pollen antigen stimulation in the presence of IL-33 compared with cells isolated from sensitized asymptomatic individuals, indicating the important role of pathogenic memory Th2 cells in the development of AR [35]. Decreased numbers of antigen-reactive ST2-positive memory T cells after SLIT was demonstrated. In particular, numbers of ST2-positive memory T cells were significantly lower in responders in SLIT group, which showed a significant correlation with the degree of improvement in nasal symptom scores [36]. A recent study using single-cell RNA sequencing and repertoire analysis using peripheral blood lymphocyte samples before and after JC pollen SLIT for JC pollinosis, showed that SLIT reduced JC pollen-specific pathogenic memory Th2 cells and induced trans-type Th2 differentiation into Tregs [37]. Therefore, the suppression of specific pathogenic Th2 cells and Th2 cell transition to Tregs may represent a central part of the mechanism of SLIT.

## 7. Conclusions

JC pollinosis is a serious problem because the number of affected people is very high in Japan. Along with unpleasant nasal symptoms, the impact on their quality of life is also significant. Recent clinical trials and clinical data of JC pollen SLIT have shown its therapeutic efficacy, safety, and long-term effects. Recent studies have demonstrated the safety of the concomitant use of JC pollen SLIT and HDM SLIT, and the therapeutic efficacy of JC pollen SLIT during the cypress pollen dispersal period following the JC pollen dispersal period. Immunological modifications induced by SLIT might be important for acquired immunity and the involvement of innate immunity. Furthermore, biomarkers of SLIT are expected to be established. JC pollen sublingual immunotherapy is currently of great significance as a fundamental treatment for JC pollinosis, which has few spontaneous remissions.

## Figures and Tables

**Figure 1 pathogens-11-01313-f001:**
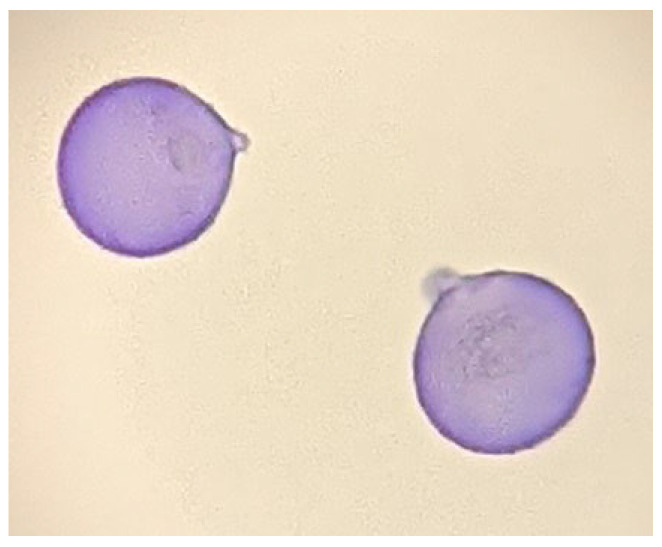
Photomicrograph image of Japanese cedar pollen collected on the rooftop of the University of Yamanashi.

**Figure 2 pathogens-11-01313-f002:**
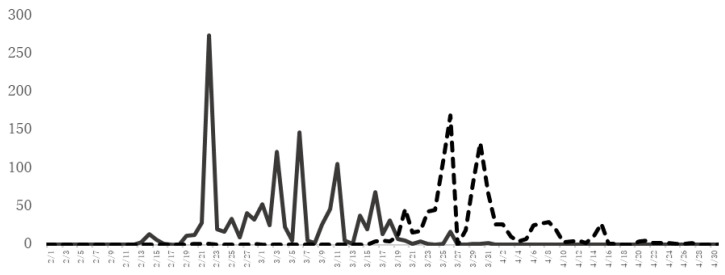
The amount of JC pollen and cypress pollen scattered. The number (grains/cm^2^) of JC pollen (solid line) and cypress pollen (dotted line) scattered as assessed by the Durham method at the University of Yamanashi, School of Medicine (Yamanashi Prefecture) from February to April 2021.

**Figure 3 pathogens-11-01313-f003:**
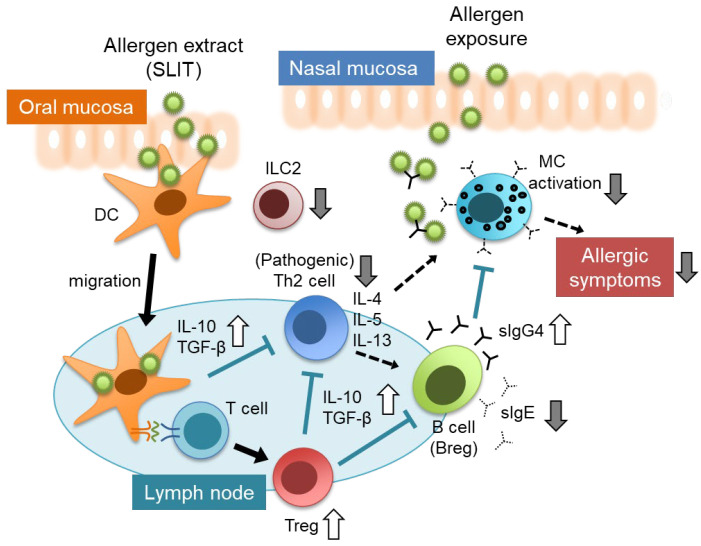
Immunological alterations induced by SLIT and putative mechanisms. DC: dendritic cell, ILC2: type 2 innate lymphoid cell, Treg: regulatory T cell, MC mast cell, sIgE: specific IgE, sIgG4: specific IgG4.

## Data Availability

Not applicable.

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
