# Peer review of "Sublingual Immunotherapy for Japanese Cedar Pollinosis: Current Clinical and Research Status"

_pathogens, 2022, doi:10.3390/pathogens11111313_

Round 1

Reviewer 1 Report

1. It  should be  better to compare similar studies for other seasonal pollen allergies in literature (i.e SLIT with Grazax and ORALAIR). 

2. It could be useful to explain the contex of SLIT with lights and shadows of this therapy (see L. Klimek, Brehler et al. Human Vaccine & immunotherapy Vol 18, 2022 Issue 5.)

3.We don't know the efficacy of oral immunotherapy in  long term. The authors could evaluate this aspect.

4. The last critical point is represented by the evaluation of immunotherapy. Symptoms score is controversial and the authors should remark  this point and to explain why they didn't adopt other laboratory parameters or clinical tests. 

Reviewer 2 Report

General comments:

The paper, titled as “Sublingual immunotherapy for Japanese cedar pollinosis: current clinical and research status”, by Daiju Sakurai et al, to review the Sublingual immunotherapy with allergic rhinitis. This article completely introduces the sublingual immunotherapy in Japan, and had many beneficial conclusions.

Specific comments:

1.       Table 1 and Table 2 are similar in content and simple in structure. Can they be combined into one table.

2.       The discussion content should also compare the differences of SCIT treatment in Japan.
